# Associations of Ultra-Processed Food Intake with the Incidence of Cardiometabolic and Mental Health Outcomes Go Beyond Specific Subgroups—The Brazilian Longitudinal Study of Adult Health

**DOI:** 10.3390/nu16244291

**Published:** 2024-12-12

**Authors:** Scheine Leite Canhada, Álvaro Vigo, Luana Giatti, Maria de Jesus Fonseca, Leidjaira Juvanhol Lopes, Letícia de Oliveira Cardoso, Carlos Augusto Monteiro, Maria Inês Schmidt, Bruce Bartholow Duncan

**Affiliations:** 1Postgraduate Program in Epidemiology, Universidade Federal do Rio Grande do Sul, Porto Alegre 90035-003, RS, Brazil; 2Department of Preventive and Social Medicine, School of Medicine & Clinical Hospital/EBSER, Universidade Federal de Minas Gerais, Belo Horizonte 30130-100, MG, Brazil; 3National School of Public Health, Fundação Oswaldo Cruz, Rio de Janeiro 21040-900, RJ, Brazil; 4Center for Epidemiological Studies in Health and Nutrition, School of Public Health, University of São Paulo, São Paulo 05508-220, SP, Brazil

**Keywords:** foods, sugar-sweetened beverages, processed meat, weight gain, cardiometabolic risk factors, epidemiology, mental disorders

## Abstract

**Background/Objectives**: Avoidance of ultra-processed foods (UPFs) has been recommended to achieve a healthy diet, but whether it applies equally to all UPFs is uncertain. We evaluated individual UPF subgroups in the prediction of cardiometabolic and mental health outcomes. **Methods**: The Brazilian Longitudinal Study of Adult Health (ELSA-Brasil) is an occupational cohort study of 15,105 adults (35–74 years) recruited in 2008–2010. We followed participants up to 2018 to ascertain elevated weight and waist gains and the onset of metabolic syndrome, hypertension, metabolic dysfunction-associated steatotic liver disease, diabetes, common mental disorders, depressive episodes, and anxiety disorders. **Results**: In adjusted robust Poisson regression, greater overall UPF intake at the baseline predicted all studied outcomes. Various subgroups of UPF, most frequently processed meat and sweetened beverages, individually conferred a greater risk, and nearly all predicted at least one studied outcome. Considering all subgroups and outcomes, a broad pattern of overall risk was present. When subgroups not individually predictive of these outcomes were aggregated, increased risk (for a one-standard deviation change) was observed for elevated weight (RR = 1.05; 95% CI 1.01–1.11) and waist (RR = 1.05; 95% CI 1.00–1.10) gains, and for the incidence of common mental (RR = 1.06; 95% CI 1.01–1.12), and anxiety (RR = 1.09; 95% CI 1.02–1.16) disorders. **Conclusions**: UPFs overall and their subgroups predicted future cardiometabolic and mental health outcomes. The pattern of individual UPF subgroup associations varied across outcomes, and the aggregate of subgroups not individually predicting risk also predicted large gains in overall and central adiposity and the incidence of mental health disorders. While additional studies investigating other outcomes are needed, these findings justify avoidance of overall UPF intake in health promotion and disease prevention.

## 1. Introduction

The consumption of ultra-processed foods (UPFs) has increased during the last four decades, starting in high-income and spreading to low- and middle-income countries [1]. According to the Nova classification system, UPFs are industrial formulations primarily or entirely made from substances extracted from foods, rich in chemical additives (colorings, flavorings, and emulsifiers), and designed to be highly convenient and appealing [2].

The recognition that UPFs were the principal food vectors leading to an excess intake of sugar, salt, fats, and multiple food additives lacking adequate risk evaluation led to the incorporation of a golden rule in the dietary guideline for Brazilian adults in 2014: “Always prefer natural or minimally processed foods and freshly made dishes and meals to ultra-processed foods” [3]. The explosion of studies demonstrating the risk of multiple diseases and death associated with greater UPF consumption has led at least 11 countries (Brazil, Uruguay, Peru, Chile, Ecuador, Mexico, Canada, France, Belgium, Israel, and India) and medical associations [4,5] to recommend reduction in UPF consumption in their nutritional interventions and population guidelines [6,7,8,9].

An umbrella review of existing meta-analyses has summarized the broad pattern of adverse health outcomes predicted by increased UPF consumption, including cardiometabolic conditions, common mental disorders, and cause-specific and all-cause mortality [10]. The hypothesis behind the NOVa classification of foods, is that the industrial process producing UPFs is harmful, and the resulting products should be avoided to the degree possible. However, over the last two years the association of UPFs with diabetes was investigated according to subgroups of UPFs, revealing that not all UPFs characterized risk [11,12,13,14]. These findings have led to different interpretations. Some argued that characterizing UPF risk in greater detail would aid consumers in adopting a healthier dietary pattern instead of the more difficult option of maximally reducing their total UPF intake [15]. Others have argued that the focus should be on the dietary pattern, not on specific foods or nutrients, thus striving to minimize overall intake of UPFs to avoid dietary risk [10].

However, given the wide range of health conditions described to date as affected by UPF intake, these issues will not be resolved by investigating one or a limited number of outcomes. That a specific UPF subgroup is not associated with one investigated outcome may not generalize to other outcomes. It is thus essential to increase the range of relevant outcomes investigated. ELSA-Brasil, a contemporary cohort assessing multiple outcomes over several years in an ethnically diverse population, offers the opportunity to contribute data to this debate by investigating the association of multiple subgroups with multiple outcomes.

To this end, we aimed to assess the association of specific UPF subgroups with the incidence of several cardiometabolic and mental health outcomes for which these products have been shown to confer risk. In these analyses our objective, rather than highlighting individual subgroup associations, was to present the overall pattern of associations and to determine whether the subgroups not individually associated with outcomes, when aggregated, would predict the future development of outcomes.

## 2. Materials and Methods

### 2.1. Study Design and Population

The Brazilian Longitudinal Study of Adult Health (in Portuguese, ‘Estudo Longitudinal de Saúde do Adulto’, ELSA-Brasil) is a multicenter prospective occupational cohort aiming to address the development and progression of chronic diseases and their risk factors, mainly focused on cardiovascular diseases and diabetes [16].

We enrolled 15,105 active or retired, non-pregnant civil servants, aged 35–74 years, from public institutions of higher education and research located in different Brazilian capital cities (Salvador, Belo Horizonte, Rio de Janeiro, São Paulo, Vitoria, and Porto Alegre) [16]. Baseline examination (Visit 1) included interviews, clinical exams, and blood sample collection conducted in research centers between August 2008 and December 2010. Participants returned to follow-up visits between 2012 and 2014 (Visit 2) and 2017–2019 (Visit 3) [17]. Starting in 2009, we also conducted annual telephone interviews to surveil outcomes. Ethics committees of each involved institution approved the research protocol, and subjects gave written consent to participate in each visit.

All measurements followed standardized protocols and regular quality control assessments. We interviewed participants using structured questionnaires to ascertain age, sex, self-declared ethnicity, educational achievement, family income, previous medical history, smoking (current and previous), alcohol consumption, physical activity in leisure time (measured using the International Physical Activity Questionnaire), family history of diseases, and medication use.

After an overnight fast, we measured weight, height, and waist circumference following internationally standardized protocols, defining body mass index (BMI) as weight (kg)/height (m)^2^. We obtained waist circumference with a 150 cm inelastic measuring tape (Mabis-Gulick, Waukegan, IL, USA) at the midpoint between the inferior edge of the costal border and the iliac crest in the mid-axillar line. We also obtained a fasting blood sample by venipuncture and conducted a standardized 2 h 75 g oral glucose tolerance test (WHO 1999). Plasma glucose was measured using hexokinase, HbA1c by high-pressure liquid chromatography, HDL-cholesterol and triglyceride levels using specific enzymatic methods, and gamma-glutamyl transferase using a kinetic colorimetric method.

We used an automatic oscillometric device to measure blood pressure three times, with an interval of one minute after five minutes of rest and averaged the last two measurements.

### 2.2. Definition of Ultra-Processed Foods

Food intake was evaluated at baseline through a previously validated food frequency questionnaire with 114 items [18]. For each item, we obtained the frequency of intake in the last year (with eight response options: ‘more than three times/day’, ‘2–3 times/day’, ‘once-daily’, ‘5–6 times/week’, ‘2–4 times/week’, ‘once/week’, ‘1–3 times/month’ and ‘never/almost never’), and the number of portions consumed, using standardized portion sizes. We then calculated the daily amount consumed for each food item in grams by multiplying its portion number, standard portion weight, and frequency. Next, we estimated the nutritional composition and energy using the University of Minnesota Nutrition Data System for Research (NDSR) software (https://www.ncc.umn.edu/products/, accessed date 1 July 2012). Finally, we attributed the 99th percentile value for a food item when that food’s value was above the 99th percentile of its distribution.

We summed the food items into three groups, according to the Nova classification, as follows: (i) non- or minimally processed foods and culinary ingredients, (ii) processed foods, and (iii) ultra-processed foods [2].

We further categorized UPFs into subgroups based on similarities in nutritional content or purpose, as conducted in a previous report [12] and detailed in the lower panel of Figure 1.

### 2.3. Outcomes

We assessed the incidence of nine categorical outcomes among those free of the condition at baseline.

Weight gain was calculated as the difference between the last visit and the baseline weight (Visit 1). We divided the weight difference by the participant’s follow-up time to obtain the annual weight gain. Annual weight gains equal to or above the 90th percentile of the sample were considered elevated. Elevated waist gain calculation followed the same approach using waist circumference measurements obtained during visits 1–3.

The criteria to define metabolic syndrome was based on the presence of at least three of the five following components: elevated fasting plasma glucose (≥100 mg/dL or use of hypoglycemic medication); elevated plasma triglyceride (≥150 mg/dL or use of fibrates or nicotinic acid); low plasma HDL cholesterol (<40 mg/dL for men and <50 mg/dL for women, or use of fibrates or nicotinic acid); elevated blood pressure (systolic blood pressure ≥ 130 mmHg, diastolic blood pressure ≥ 85 mmHg, or confirmed use of antihypertensive medication); and abdominal obesity (waist circumference ≥ 94 cm for men and ≥80 cm for women) [19].

We considered hypertension to be present if at least one of three criteria was met: systolic blood pressure ≥ 140 mmHg, diastolic blood pressure ≥ 90 mmHg, or use of antihypertensive medication in the two weeks before the interview.

The definition of metabolic dysfunction-associated steatotic liver disease (MASLD) included the presence of hepatic steatosis accompanied by at least one of five cardiometabolic risk factors [20]. For this analysis, we defined hepatic steatosis based on a Fatty Liver Index (FLI) > 60 [21]. These five cardiometabolic risk factors were as follows: (1) overweight/obesity (BMI ≥ 25 kg/m^2^ or ≥23 kg/m^2^ for Asians, or waist circumference ˃ 94 cm for men and >80 cm for women); (2) presence of intermediate hyperglycemia (fasting plasma glucose ≥ 100 mg/dL, two-hour post-load glucose ≥ 140 mg/dL or HbA1c ≥ 5.7%) or type 2 diabetes; (3) elevated blood pressure (≥130/85 mmHg) or antihypertensive treatment; (4) elevated triglycerides (≥150 mg/dL) or lipid-lowering treatment; or (5) low HDL-cholesterol (≤40 mg/dL for men and ≤50 mg/dL for women) or lipid-lowering treatment.

Type 2 diabetes ascertainment was based on self-report and laboratory measurements obtained during the visits and on information from the annual telephone interviews. We ascertained cases based on the presence of any of the following criteria: (1) self-reported medical diagnosis of diabetes or current use of medication for diabetes; (2) laboratory measurements reaching the thresholds for fasting plasma glucose (≥126 mg/dL), two-hour post-load glucose (≥200 mg/dL), or HbA1c (≥6.5%) [22]; or (3) report of a diagnosis of diabetes on at least two annual telephone interviews after the last clinic visit.

We applied the Clinical Interview Schedule-Revised (CIS-R) to define three psychiatric outcomes measured during Visits 1 and 3: common mental disorders (CIS-R score equal to or higher than 12); depressive episodes (including all types and severities); and anxiety disorders (including general anxiety, social anxiety, panic, phobias, and obsessive-compulsive disorders) [23]. For depressive episodes we also used data collected during Visit 2.

For participants who did not attend Visit 3, we used the data obtained in Visit 2 for all outcomes except common mental disorders and anxiety disorders, as information for these was not collected at Visit 2.

### 2.4. Statistical Analysis

We describe participant characteristics and outcomes using absolute and relative frequencies for categorical variables and mean (standard deviation) or median (25th–75th percentiles) for continuous variables. We evaluated UPF intake in g/day and expressed results for a one-standard deviation difference in UPF intake. We chose to use grams instead of kcal to express the quantity of UPF because some foods and beverages do not provide energy.

Using robust Poisson regression, we first investigated associations of UPFs with each specific outcome, estimating crude and adjusted relative risks (RRs) and 95% confidence intervals (95% CI). We adjusted for age (in years), sex (male or female), race/color (white, brown, black, Asian, or Indigenous), school achievement (less than elementary, elementary, secondary, or college/university), and per capita family income (in Brazilian currency, reais). In addition, we included risk factors such as smoking (never, former or current), leisure time physical activity (in MET- minutes/week), and alcohol consumption (in grams/week). We did not adjust for additional risk factors (e.g., hypertension and BMI) that could mediate the associations in question. Next, we investigated associations of prespecified UPF subgroups with each outcome when all subgroups were present simultaneously in the models. Subgroups not found to be individually associated with an outcome were then aggregated and analyzed together to assess whether conjointly they conferred increased risk. To produce these UPF aggregates, we removed the intake of individually associated UPF subgroups from their total, and in further analyses, we included them as covariates in the fully adjusted model. We assessed multicollinearity between variables, setting a limit of 2 for the variance inflation factor.

As an additional analysis, we also included energy intake (in kcal/day) as an adjustment variable in the models where total UPF intake and its subgroups were the exposures (Appendix A). We conducted all analyses using the statistical software package SAS Studio^®^ 3.8 (SAS OnDemand for Academics) and defined an alpha value of <0.05 to denote statistical significance.

## 3. Results

The definition of the analytic sample is shown in Appendix A. Of the 15,105 participants, we initially excluded those with implausible food intake (<600 kcal/d or >6000 kcal/d), missing data on covariates, who died or were lost to follow-up before Visit 2, leaving 13,615 participants available for analyses. We then proceeded with specific exclusions for each outcome, such as prevalent cases, missing values, and those who died/were lost after Visit 2 without providing information on the outcome of interest. After these additional exclusions, our analytical sample consisted of 13,316 participants for elevated weight gain, 13,297 for elevated waist gain, 8115 for metabolic syndrome, 8744 for hypertension, 8699 for MASLD, 11,405 for diabetes, 8745 for common mental disorders, 12,753 for depressive episodes, and 9971 for anxiety disorders.

Table 1 shows the baseline characteristics of the study sample before the specific exclusions referent to each outcome. For this sample, the mean (standard deviation, SD) age was 52.0 (9.0) years; there were slightly more women (55.2%) than men, with approximately half (52.7%) being Whites and the majority (53.8%) having at least a university degree. The mean intake of UPF was 436 (293) grams/day, most resulting from the intake of sweetened beverages, non-dairy sweet snacks and desserts, yogurt and dairy sweets, and ready-packaged bread.

As seen in Table 2 (left), over a follow-up of approximately eight years for all outcomes, incidence varied from 10.0% to 31.3% for the cardiometabolic risk factors and diseases and from 3.1% to 12.9% for the mental health outcomes. On the right side of this table, we see that the intake of UPFs was positively associated with all outcomes, with relative risks for a one SD increase in intake ranging from 1.06 (95% CI 1.03–1.10; hypertension) to 1.30 (95% CI 1.20–1.41; depressive episodes).

As seen in Table 3, a positive association, here expressed for a one SD increase in intake, was present with at least one individual UPF subgroup across all outcomes. For both elevated weight and waist gains, the relative risk for processed meats was 1.07 (95% CI 1.02–1.13), and for distilled alcoholic beverages, 1.09 (95% CI 1.04–1.15) for elevated weight gain and 1.14 (95% CI 1.08–1.20) for elevated waist gain. For metabolic syndrome and hypertension, sweetened beverages had RRs of 1.09 (95% CI 1.06–1.12) and 1.08 (95% CI 1.04–1.12), respectively. For MASLD, baked and fried snacks, processed meats, and sweetened beverages were associated with corresponding relative risks of 1.05 (95% CI 1.01–1.09), 1.09 (95% CI 1.05–1.13), and 1.09 (95% CI 1.05–1.12), respectively. For diabetes, processed meats and sweetened beverages were associated, their relative risks being 1.07 (95% CI 1.03–1.12) and 1.14 (95% CI 1.09–1.18), respectively. For common mental disorders, non-dairy sweet snacks and desserts, ready-to-eat/heat-mixed dishes, and sweetened beverages were associated with RRs of 1.08 (95% CI 1.02–1.13), 1.07 (95% CI 1.01–1.12), and 1.10 (95% CI 1.05–1.16), respectively. For depressive episodes, results were similar, with a higher risk associated with non-dairy sweet snacks and desserts (RR 1.13, 95% CI 1.03–1.22) and sweetened beverages (RRs 1.23, 95% CI 1.13–1.34). For anxiety disorders, ready-packaged bread and processed meats individually elevated the risk with RRs of 1.10 (95% CI 1.03–1.17) and 1.08 (95% CI 1.01–1.15), respectively. All associations, including those without statistical significance, are presented in Appendix A.

Figure 2 summarizes the pattern of UPF subgroups associations in the form of a heat map. The colors red and green represented risk and protection, respectively, with darker red and green indicating statistical significance. The first column shows the percentage of the daily UPF intake of the nine subgroups. Sweetened beverages (34.5%), ready-packaged bread (16.6%), non-dairy sweet snacks and desserts (14.5%), and yogurt and dairy sweets (14.1%) comprised the bulk (79.7%) of UPF intake in grams.

This approach, presenting both significant associations and tendencies toward risk or protection, allows a comprehensive overview of the associations found. First, it shows a marked variation in subgroups presenting statistically significant risk across the nine outcomes. While sugary drinks and processed meat stand out, all but two individual subgroups presented a statistically significant risk for at least one outcome. One without associations, spreads, comprised only 2.6% of the daily UPF intake. Additionally, we found two statistically significant protective associations. Greater intake of ready-packaged bread, despite presenting risk for anxiety disorders, was protective against metabolic syndrome. Greater intake of yogurt and dairy sweets, despite tending toward risk for most other outcomes, conferred protection against diabetes. Second, while most comparisons lacked statistical significance, the overall pattern indicated risk, as red colors were approximately four times more frequent than green ones, and almost all significant associations indicated risk.

Figure 3 shows that, after the removal of UPF intake in subgroups that individually conferred risk, the aggregate of remaining UPF intake, when analyzed in models which additionally adjusted for the individual subgroup associations, continued to confer risk for elevated weight (RR 1.05, 95% CI 1.01–1.11) and waist (RR 1.05, 95% CI 1.00–1.10) gains, common mental disorders (RR 1.06, 95% CI 1.01–1.12), and anxiety disorders (RR 1.09, 95% CI 1.02–1.16). For diabetes, the remaining aggregate was protective (RR 0.94; 95% CI 0.89–0.98).

## 4. Discussion

In our sample of middle-aged and elderly Brazilians, overall UPF intake predicted the risk of developing nine cardiometabolic and mental health disorders. Most subgroups of UPF predicted risk for at least one outcome, with sugary beverages and processed meat being more prominent. Subgroups not conferring risk individually conferred a greater risk for large weight and waist gains and for two mental health outcomes when aggregated.

When analyzed overall, our findings that UPFs increased risk for all evaluated outcomes is consistent with recent estimates of an umbrella review summarizing risk for a wide array of conditions, including those examined here [10].

Previous analyses of UPF subgroups have been conducted primarily with regard to diabetes. One study based on three US cohorts of health professionals [11] showed greater incidence of diabetes with increased intake of refined breads; sauces, spreads, and condiments; artificially and sugar-sweetened beverages; animal-based products; and ready-to-eat mixed dishes, and decreased incidence with greater intake of cereals; dark and whole-grain breads; packaged sweet and savory snacks; fruit-based products; and yogurt and dairy-based desserts. An ELSA-Brasil study found a similar pattern of UPF subgroups to be related to diabetes [12]. The ARIC Study [13] also found greater diabetes risk with increased intake of sugar and artificially sweetened beverages and ultra-processed meat, and, in addition, for sugary snacks. The Korean Genome and Epidemiology Study Ansan-Ansung Cohort [14] found independent diabetes risk with greater consumption of ham/sausage, instant noodles, ice cream, and carbonated beverages.

UPF subgroups analyses with other outcomes evaluated frailty and death. In the Nurses’ Health Study, a greater incidence of frailty was found with increased intake of artificial and sugar-sweetened beverages; fat, spreads, and condiments; and yogurt and dairy-based desserts [24]. In the EPIC-NL Study, ultra-processed beverages but not ultra-processed foods were associated with greater mortality [25]. That a meta-analysis of refined grains suggests a lesser risk than generally assumed for this food group increases the importance of better-characterizing dietary risk factors for diabetes [26].

A basic assumption underlying our investigation is that a healthy eating recommendation should be based on the prevention of many health conditions causing major disease burden. In this context, our finding that seven of the nine subgroups conferred increased risk with at least one outcome (sweetened beverages with six; processed meat with five; distilled alcoholic beverages, and non-dairy sweet snacks and desserts with two; and baked and fried snacks, ready-to-eat/heat-mixed dishes, and ready-packaged bread with one outcome each), strengthens the evidence that most UPF subgroups are involved in risk to overall health.

Additional support for the thesis that UPFs should be considered overall is that even after excluding individually predictive UPF subgroups, the remaining aggregate of UPFs still conferred an increased risk for overall and central adiposity, which are among the major drivers of today’s chronic disease burden, as well as for prevalent mental health outcomes which also produce a major burden. This finding supports the contention that the risk resulting from greater UPF intake probably arises not only from specific food items or nutrients but also from additional, still not entirely understood mechanisms resulting from the ultra-processing of foods.

Numerous reasons for the effects of UPFs have been proposed. In addition to increasing weight and waist [27], UPFs confer poor nutritional quality, with low quantities of vitamins and fibers and high amounts of energy, sugar, salt, and fats [28]. Components of UPFs such as food additives (artificial sweeteners, emulsifiers, for example), substances formed during the industrial processes (furans, heterocyclic amines, acrolein, acrylamide, and advanced glycation end products), and packaging contact material (bisphenols and phthalates), appear to have harmful effects on the human body [29,30,31,32]. Among the potential mechanisms of action of these components are interference with cell signaling pathways involved in glucose homeostasis [32] and alterations in the gut microbiota and gut barrier function, which can lead to inflammation and metabolic changes [33,34]. Intensive processing may also affect digestion, nutrient absorption, and satiety by altering the food matrix [35]. Additionally, the addictive potential of combined high loads of fat and rapidly absorbed carbohydrates may lead to addictive eating behavior, producing both greater adiposity, and mental health disorders [36].

Possible limitations to our study deserve note. First, food frequency questionnaires are known to present problems with precision and biased responses. Further, our food frequency questionnaire was not explicitly designed to evaluate the Nova classification groups, which may have led to an underestimation of the risk in some of the associations reported. An excellent example of this difficulty in classification is yogurt, which, while in its basic form has been suggested to protect against cardiometabolic diseases, is considered here a UFP, given that most yogurt consumed by our participants contained sugar and other additives. Although errors in classification could be present, the frequency of UPF intake based on the ELSA-Brasil cohort questionnaire at baseline was similar to what was found in a nationally representative survey [37]. Second, although we adjusted for multiple potential confounders, residual or unmeasured confounding cannot be ruled out in our investigation of the pattern of UPF associations across multiple outcomes. However, the independent associations we found of overall UPFs with the various outcomes are amply supported by the literature [10]. Third, evaluating multiple UPF subgroups across nine outcomes requires multiple statistical tests which may produce spurious associations. However, the broader pattern of risk for UPFs seen with greater clarity when both significant and non-significant subgroup findings are displayed, is unlikely to be caused by chance alone. Our hypothesis of associations being present after removing individually associated subgroups involved fewer tests, and the results of these were frequently significant. Nonetheless, the risk of false positive associations is possible.

Our study has significant strengths. ELSA-Brasil is an ethnically diverse cohort study with an appreciable sample size and minor losses to follow-up. Being enrolled in a contemporary cohort, our participants´ nutritional intake more closely reflects a current scenario of UPF intake. We performed highly standardized measurements with strict quality control, including applying a validated questionnaire to assess diet. Finally, the breadth of ELSA´s characterization of chronic diseases and their risk factors permitted evaluation in the same population of many conditions highly relevant to population health.

Finally, our findings support nutritional guidance based on avoiding the whole spectrum of UPFs to prevent cardiometabolic and mental health outcomes. Such guidance will also favorably impact planetary health [38,39].

## 5. Conclusions

In conclusion, we confirmed positive associations between overall UPF intake and multiple chronic conditions. That many and varied UPF subgroups individually predicted the development of specific outcomes produced an overall picture of UPF intake, viewed at the level of its subgroups, as one of risk. Moreover, after removing these subgroups, remaining UPFs still predicted a greater incidence of important outcomes. These findings add to the mounting evidence that the construct of UPFs is valid and not an overly broad classification in which only some elements are relevant. They support the contention that clinical counseling and public health recommendations and messaging related to nutrition should include the avoidance of overall UPF intake.

## Figures and Tables

**Figure 1 nutrients-16-04291-f001:**
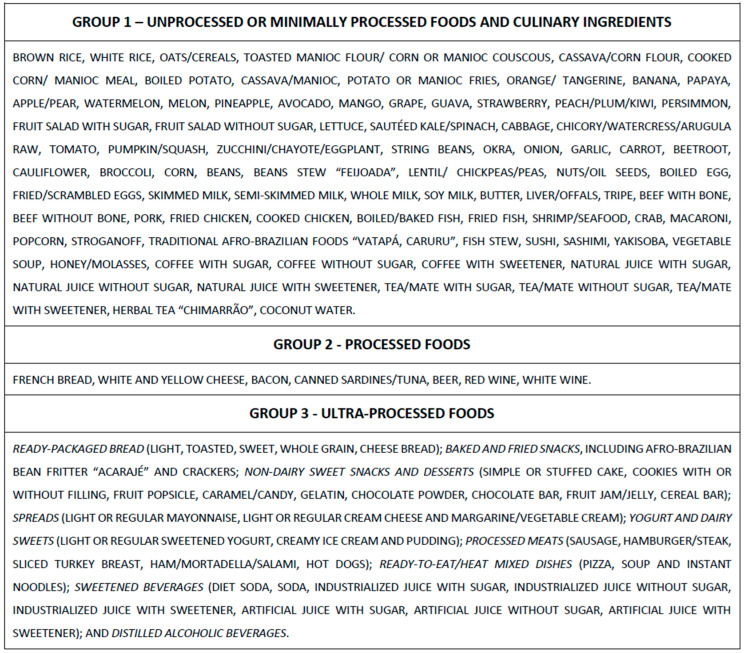
ELSA-Brasil food groups, according to the Nova classification, are based on the degree of industrial processing of food. The ultra-processed food group (Group 3) was divided into prespecified subgroups based on similarities in nutritional content or purpose.

**Figure 2 nutrients-16-04291-f002:**
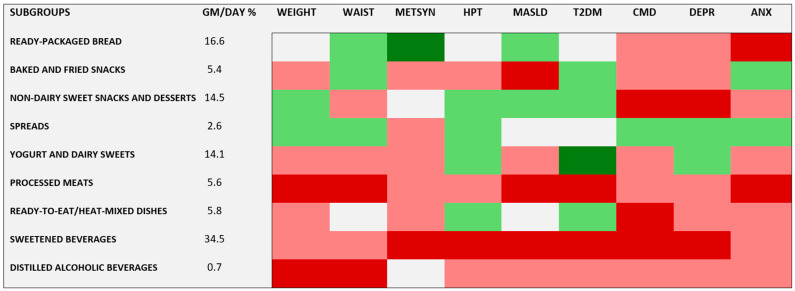
Heat map of the association of UPF subgroups and the nine outcomes. Red = risk (RR > 1); Green = protection (RR < 1); White = no association (RR = 1); Darker colors = statistical significance; GM/DAY % = Percent of total dietary intake of UPFs in grams/day; WEIGHT = elevated weight gain; WAIST = elevated waist gain; METSYN = metabolic syndrome; HPT = hypertension; MASLD = metabolic dysfunction-associated steatotic liver disease; T2DM = diabetes; CMD = common mental disorders; DEPR = depressive episodes; and ANX = anxiety disorders.

**Figure 3 nutrients-16-04291-f003:**
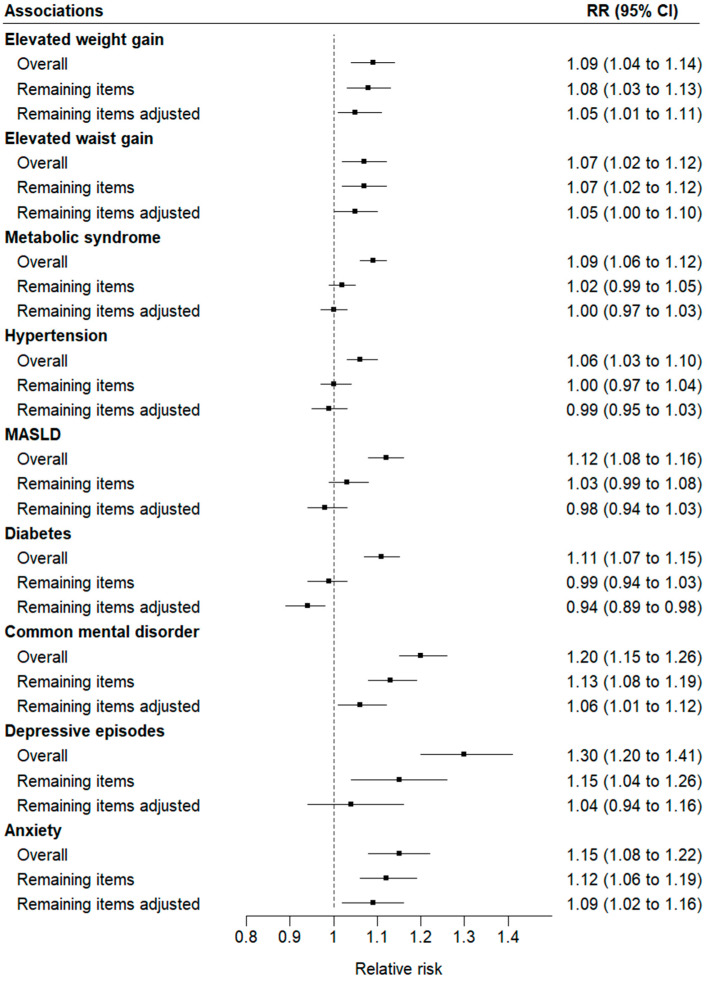
Associations of cardiometabolic and mental health outcomes for a one-standard deviation difference in (1) overall ultra-processed food (UPF) intake; (2) the aggregate of UPF intake remaining after removal of UPF subgroups presenting statistically significant associations; and (3) this aggregate, now additionally adjusted for the individually associated subgroups. Analyses are conducted independently for each outcome. All models are adjusted for age, sex, race/color, school achievement, per capita family income, smoking, physical activity, and alcohol consumption using robust Poisson regression.

**Table 1 nutrients-16-04291-t001:** Baseline characteristics of the overall study sample before specific exclusion for each outcome. ELSA-Brasil (n = 13,651).

Characteristics	Total (n = 13,651)					
	Mean	SD	N	%	Median	IQR
Age (years)	52.0	9.0				
Sex						
Female			7528	55.2		
Race/skin color						
Black			2172	15.9		
Brown			3801	27.8		
White			7191	52.7		
Asian			343	2.5		
Indigenous			144	1.1		
Income (reais)					1411	726–2282
Education						
Less than elementary			732	5.3		
Elementary			887	6.5		
Secondary			4691	34.4		
College/university			7341	53.8		
Smoking						
Never			7888	57.8		
Former smoker			4076	29.9		
Current smoker			1687	12.3		
Physical activity (MET-minutes/week)					240	0–960
Alcohol consumption (g/d)					0	0–67.2
Energy intake (kcal/d)	2613	932				
UPF (g/d)	436	293				
UPF subgroups (g/d)						
Ready-packaged bread	56.2	48.0				
Baked and fried snacks	19.0	21.0				
Non-dairy sweet snacks and desserts	57.4	57.0				
Spreads	9.3	11.6				
Yogurt and dairy sweets	56.5	61.9				
Processed meats	20.9	21.6				
Ready-to-eat/heat-mixed dishes	21.8	25.2				
Sweetened beverages	192	231				
Distilled alcoholic beverages	2.3	8.4				

**Table 2 nutrients-16-04291-t002:** Total sample, incidence, follow-up time, and overall associations for a one-standard deviation difference in total ultra-processed food intake for each outcome. ELSA-Brasil.

	Total	Incident		Follow-Up	Difference in One-Standard Deviation **	
	Sample	Cases		Time *	Crude	Adjusted ***
Outcomes	N	N	%	Years	RR (95% CI)	RR (95% CI)
Elevated weight gain	13,316	1333	10.0	7.8 (1.3)	1.17 (1.12–1.22)	1.09 (1.04–1.14)
Elevated waist gain	13,297	1376	10.4	7.8 (1.4)	1.12 (1.08–1.17)	1.07 (1.02–1.12)
Metabolic syndrome	8115	2541	31.3	7.9 (1.3)	1.10 (1.07–1.13)	1.09 (1.06–1.12)
Hypertension	8744	2013	23.0	7.9 (1.3)	1.03 (0.99–1.07)	1.06 (1.03–1.10)
MASLD	8699	1893	21.8	7.9 (1.3)	1.17 (1.13–1.20)	1.12 (1.08–1.16)
Diabetes	11,405	1884	16.5	7.8 (1.3)	1.08 (1.04–1.12)	1.11 (1.07–1.15)
Common mental disorders	8745	1128	12.9	8.2 (0.5)	1.20 (1.15–1.25)	1.20 (1.15–1.26)
Depressive episodes	12,753	393	3.1	7.8 (1.3)	1.28 (1.19–1.39)	1.30 (1.20–1.41)
Anxiety disorders	9971	789	7.9	8.2 (0.5)	1.15 (1.08–1.21)	1.15 (1.08–1.22)

* Expressed as mean (standard deviation). ** SD of UPF intake: 292 g/d for weight and waist gain, 281 g/d for metabolic syndrome, 289 g/d for hypertension, 271 g/d for MASLD, 291 g/d for diabetes, 281 g/d for common mental disorders, 290 g/d for depressive episodes, and 286 g/d for anxiety disorders. *** Adjusted through robust Poisson regression for age, sex, race/color, school achievement, per capita family income, smoking, physical activity, and alcohol consumption.

**Table 3 nutrients-16-04291-t003:** Statistically significant associations of ultra-processed food (UPF) subgroups with nine cardiometabolic and mental health outcomes, for a one-standard deviation increase in intake *. ELSA-Brasil.

	Crude	Adjusted **
Outcomes	RR (95% CI)	RR (95% CI)
Elevated weight gain		
Processed meats	1.11 (1.06–1.16)	1.07 (1.02–1 13)
Distilled alcoholic beverages	1.00 (0.95–1.05)	1.09 (1.04–1.15)
Elevated waist gain		
Processed meats	1.08 (1.03–1.14)	1.07 (1.02–1.13)
Distilled alcoholic beverages	1.01 (0.96–1.06)	1.14 (1.08–1.20)
Metabolic syndrome		
Sweetened beverages	1.10 (1.07–1.14)	1.09 (1.06–1.12)
Hypertension		
Sweetened beverages	1.08 (1.04–1.12)	1.08 (1.04–1.12)
MASLD		
Baked and fried snacks	1.07 (1.03–1.11)	1.05 (1.01–1.09)
Processed meats	1.13 (1.09–1.17)	1.09 (1.05–1.13)
Sweetened beverages	1.12 (1.08–1.16)	1.09 (1.05–1.12)
Diabetes		
Processed meats	1.05 (1.00–1.09)	1.07 (1.03–1.12)
Sweetened beverages	1.14 (1.09–1.18)	1.14 (1.09–1.18)
Common mental disorders		
Non-dairy sweet snacks/desserts	1.10 (1.05–1.15)	1.08 (1.02–1.13)
Ready-to-eat/heat-mixed dishes	1.03 (0.97–1.08)	1.07 (1.01–1.12)
Sweetened beverages	1.10 (1.05–1.16)	1.10 (1.05–1.16)
Depressive episodes		
Non-dairy sweet snacks/desserts	1.17 (1.08–1.28)	1.13 (1.03–1.22)
Sweetened beverages	1.23 (1.13–1.33)	1.23 (1.13–1.34)
Anxiety disorders		
Ready-packaged bread	1.07 (1.00–1.15)	1.10 (1.03–1.17)
Processed meats	1.06 (0.99–1.13)	1.08 (1.01–1.15)

* One-standard deviation differences for UPF subgroups varied slightly across outcomes due to differences in exclusions. For the associations found, they were 47.5 g/d for ready-packaged bread, 19.9 g/d for baked and fried snacks, ranged from 54.1 to 56.2 g/d for non-dairy sweet snacks and desserts, from 20.3 to 21.6 g/d for processed meat, 24.4 g/d for ready-to-eat/heat-mixed dishes, from 208 to 228 g/d for sweetened beverages, and 8.4 g/d for distilled alcoholic beverages. ** Adjusted in robust Poisson regression for age, sex, race/color, school achievement, per capita family income, smoking, physical activity, and alcohol consumption.

## Data Availability

Considering guidelines placed by the ethics committees responsible for ELSA’s study centers, the data used in this study can be made available by request to ELSA’s Publications Committee (pal@ups.br). Additional information can be obtained from the ELSA Coordinator from the Research Center of Rio Grande do Sul (maria.schmidt@ufrgs.br).

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
