# Peer review of "Associations of Ultra-Processed Food Intake with the Incidence of Cardiometabolic and Mental Health Outcomes Go Beyond Specific Subgroups—The Brazilian Longitudinal Study of Adult Health"

_nutrients, 2024, doi:10.3390/nu16244291_

Round 1
Reviewer 1 Report
Comments and Suggestions for Authors
The authors assessed the use of highly processed food on cardiovascular health and broadly understood cognitive processes. The topic is interesting and important. I have a few comments on the work and I would like to ask you to respond to them:
- the arbitrarily adopted blood pressure values are different from those presented by the AHA or ESC (in 2024). I understand that the values were adopted before the study began, hence such values. I believe that reference should be made to the latest AHA and ESC/EHS guidelines and both the goals and values should be explained
- physical activity was expressed in MET-minutes/week. Is it possible to specify this effort more precisely, its distribution during the week, or was it sometimes a few hours of activity once a week? What was its type? Aerobic, strength?
- "Yogurt and dairy sweets" is a UPF group... but you can make excellent yogurt at home, it's not fat and it's absolutely sugar-free... it's best to clarify this group
- please correct the application to make it more explicit. If something is harmful, then it's probably necessary to clearly suggest the need for immediate education and changing eating habits
Author Response
Comment 1: the arbitrarily adopted blood pressure values are different from those presented by the AHA or ESC (in 2024). I understand that the values were adopted before the study began, hence such values. I believe that reference should be made to the latest AHA and ESC/EHS guidelines and both the goals and values should be explained
Response:
Thank you for your suggestion. In Brazil, the cutoff points for diagnosing hypertension are defined by the Brazilian Society of Cardiology, and to date remains the same as those described in our article (Barroso et al. Brazilian Guidelines of Hypertension - 2020. Arq Bras Cardiol. 2021;116:516–658. doi: 10.36660/abc.20201238).
Comment 2: physical activity was expressed in MET-minutes/week. Is it possible to specify this effort more precisely, its distribution during the week, or was it sometimes a few hours of activity once a week? What was its type? Aerobic, strength?
Response:
Physical activity was measured using the International Physical Activity Questionnaire (IPAQ) section on leisure activities, which considers minutes of activity in light, moderate and vigorous activities, which it defines to include walking, running, gym exercises, swimming, among others. The questionnaire provides no details related to timing or type of activity.
We have now detailed more this in the methods section: “...physical activity in leisure time (measured using the International Physical Activity Questionnaire), ...”.
The distribution of this measure can be seen in table 1.
Comment 3: "Yogurt and dairy sweets" is a UPF group... but you can make excellent yogurt at home, it's not fat and it's absolutely sugar-free... it's best to clarify this group
Response:
We agree, although the yogurt you describe is not an ultra-processed food. The majority of yogurt consumed in Brazil is being sold containing, in addition to sugar, various additives. As our food frequency questionnaire, like those of most cohorts reporting UPF associations, does not distinguish between these types of yogurt, we, like most others, opted to consider yogurt as an ultra-processed food.
We have added text to the limitations section about this:
“…Nova classification groups, which may have led to an underestimation of the risk in some of the associations reported. An excellent example of this difficulty in classification is yogurt, which, while in its basic form has been suggested to protect against cardiometabolic diseases, is considered here a UFP, given that most yogurt consumed by our participants contained sugar and other additives.”
Comment 4: please correct the application to make it more explicit. If something is harmful, then it's probably necessary to clearly suggest the need for immediate education and changing eating habits
Response:
We believe that nutritional recommendations are best made by convened groups of experts based on evidence gathered across multiple studies. Our role here, as investigators, is rather to furnish that evidence, thus providing subsidies for the work of these groups.
Reviewer 2 Report
Comments and Suggestions for Authors
This study was aimed to assess the association of specific subgroups of ultra-processed foods (UPFs) with the incidence of several cardiometabolic and mental health outcomes for which these products have been shown to confer risk. As the results, UPFs overall and their subgroups predicted future cardiometabolic and mental health outcomes. The reviewer believes that the results of this study highlight that UPF intake is associated with cardiometabolic and mental health outcomes and that the results of present study provide very useful information from the perspective of preventing the development of future cardiovascular disease. Additionally, the reviewer also thinks that the present study is well designed and the analysis is appropriately conducted. However, there are several questions in this study.
1. For the novelty of this study, as explained in the "Introduction" section, an umbrella review of existing meta-analyses has summarized the broad pattern of adverse health outcomes predicted by increased UPF consumption, including cardiometabolic conditions, common mental disorders, and cause-specific and all-cause mortality. In this study, the authors subdivided outcome categories, but these were simply traditional risk factors such as obesity, hypertension, and diabetes. The authors should mention the novelty and clinical implications for preventive medicine that this study differs from previous studies.
2. As mentioned above, there are reports of an association of UPF intake with the incidence and progression of cardiometabolic diseases and mental disorders. The most important goal of cardiometabolic disease prevention is to prevent the incidence and progression of atherosclerotic cardiovascular disease. The reviewer considers that it is more novel to examine the association between CPF intake and the pathology of various cardiovascular diseases, in addition to traditional cardiometabolic risk factors.
3. For the primary outcome in this study, it is difficult to understand what the authors want to emphasize in this study because this study has many outcomes and the primary outcome is not clear. Authors should clearly state the primary outcome of their study and develop their argument based on the primary outcome.
4. In this study, the authors present the results of UPF and UPF subgroups in gram terms. The nutritional data generally uses data adjusted by the density method because these data are affected by total energy intake. Did the authors make any adjustments to the UPF and UPF subgroup data in this study?
5. In Table 3, the authors present only the data that were statistically significant. The reviewer thinks that the results of the UPF subgroups that were not statistically significant for each outcome should also be presented.
6. In relation to the above, the reviewer infers that Figure 2 reflects the results of Table 3. The reviewer thinks that it is acceptable to combine the results in Figure 2 and Table 3 if they show the same results. For example, add the numerical information in Table 3 to Figure 2.
Author Response
Comment 1: For the novelty of this study, as explained in the "Introduction" section, an umbrella review of existing meta-analyses has summarized the broad pattern of adverse health outcomes predicted by increased UPF consumption, including cardiometabolic conditions, common mental disorders, and cause-specific and all-cause mortality. In this study, the authors subdivided outcome categories, but these were simply traditional risk factors such as obesity, hypertension, and diabetes. The authors should mention the novelty and clinical implications for preventive medicine that this study differs from previous studies.
Response:
A relevant question today is whether UPFs as an overall group are harmful, or whether only some UPF subgroups cause harm. The novelty of this report is its investigation of associations of UPF sub-groups with multiple outcomes. We aimed to characterize the extent to which risk of these subgroups varies across outcomes and whether remaining UPFs, once individually associated sub-group consumption was removed, continue to predict risk.
We have altered the next to last paragraph of the introduction to now read: “…ELSA-Brasil, a contemporary cohort assessing multiple outcomes over several years in an ethnically diverse population, offers the opportunity to contribute data to this debate by investigating the association of multiple subgroups with multiple outcomes.”
Comment 2: As mentioned above, there are reports of an association of UPF intake with the incidence and progression of cardiometabolic diseases and mental disorders. The most important goal of cardiometabolic disease prevention is to prevent the incidence and progression of atherosclerotic cardiovascular disease. The reviewer considers that it is more novel to examine the association between CPF intake and the pathology of various cardiovascular diseases, in addition to traditional cardiometabolic risk factors.
Response:
Thank you for your comment. We selected these nine outcomes because our aim was to address cardiometabolic and mental health conditions and risk factors which were assessed during baseline and at least one follow-up visit of the participants to the research centers and for which statistical power exists in ELSA, as shown by previous analyses. Unfortunately, although our study does capture cardiovascular outcomes such as myocardial infarction, stroke, and CVD deaths, our follow-up for these rarer outcomes has yet to produce a sufficient number of cases to permit analyses with adequate statistical power.
Comment 3: For the primary outcome in this study, it is difficult to understand what the authors want to emphasize in this study because this study has many outcomes and the primary outcome is not clear. Authors should clearly state the primary outcome of their study and develop their argument based on the primary outcome.
Response:
We have rewritten the final paragraph of the introduction:
“To this end, we aimed to assess the association of specific UPF subgroups with the incidence of several cardiometabolic and mental health outcomes for which UPFs have been shown to confer risk. In these analyses our objective, rather than highlighting individual UPF subgroup associations, was to present the overall pattern of these associations and to determine whether subgroups not individually associated with outcomes, when aggregated, would predict the future development of outcomes.”
Comment 4: In this study, the authors present the results of UPF and UPF subgroups in gram terms. The nutritional data generally uses data adjusted by the density method because these data are affected by total energy intake. Did the authors make any adjustments to the UPF and UPF subgroup data in this study?
Response:
We tested the inclusion of total energy intake as an adjustment variable in our analyses, both in the analyses of ultra-processed foods as a group and in the analyses of individual subgroups. The estimates are quite similar, with the inclusion of energy increasing some relative risks. We chose not to present these models because energy may be a potential mediator of the associations under study.
We have added a sentence to the Statistical Analysis section:
“As an additional analysis, we also included energy intake (in kcal/day) as an adjustment variable in the models where total UPF intake and its subgroups were the exposures (Supplementary Table S1)”.
Comment 5: In Table 3, the authors present only the data that were statistically significant. The reviewer thinks that the results of the UPF subgroups that were not statistically significant for each outcome should also be presented.
Response:
Unfortunately, adding all these data in Table 3 would make it hard for readers to grasp the important associations. However, we have now included these results you suggested in the supplementary file and added the following sentence to the Results section: “All associations, including those without statistical significance, are presented in Supplementary Table S1”.
Comment 6: In relation to the above, the reviewer infers that Figure 2 reflects the results of Table 3. The reviewer thinks that it is acceptable to combine the results in Figure 2 and Table 3 if they show the same results. For example, add the numerical information in Table 3 to Figure 2.
Response:
Thank you for your suggestion. Our idea of Figure 2 was to highlight the overall pattern of risk, signaling associations that were statistically significant and those which were only suggestive of risk or protection. Since the presentation of the findings for 81 associations is already complex for a reader to grasp, we prefer to provide the corresponding quantitative information in a separate table. This also allows the presentation of both crude and adjusted values in Table 3.
Reviewer 3 Report
Comments and Suggestions for Authors
Association of ultra-processed food intake with incidence of cardiometabolic and health outcomes go beyond specific subgroups—the Brazilian Longitudinal Study of Adult Health, Canhada et al.
This large, prospective cohort study offers unique insights into the potential risk of persistent intakes of UPF and various health conditions. By categorizing the NOVA UPF classification into food groups based on nutritional content and purpose, the researchers evaluated risk based on subtype of UPF—this novel approach contributes unique findings to the field. The ethnically diverse sample allows for generalization of the findings to the Brazilian population, excepting pregnant women since they were excluded from the cohort. Specific suggestions follow.
Keywords
Suggest adding processed meat
Introduction
Good overview of topic and introduction to study aim.
Materials and Methods
Are the structured questionnaire (line 96) and CIS-R (line 100) validated tools? If so, note that. If not, add as limitation that these tools were not validated.
Employment of Poisson appropriate given aim to evaluate probability of event given fixed interval of exposure.
Results
Handling of missing data was addressed.
Table 1: Suggest separating the first column into three discrete columns: (1) N, (2) mean, and (3) median. Re alcohol consumption, why is it 0 g/day but distilled alcoholic beverages 2.3 g/d?
Discussion
Comparison with US cohorts (line 332): Other studies have found that intake of refined grains is not associated with increased risk of T2D when you consider refined grains as a distinct food category. I think this is an important point to make given the study aim. See 10.1016/j.mayocp.2022.05.004.
Numerous reasons for harmful effects (line 361): Recommend including information on theories of how substances in UPFs (additive, preservatives, artificial colors/flavors) may disrupt human physiology. For example, leaky gut hypothesis, induction of Maillard Reaction theory, and cellular inflammation.
Limitations (lines 371-372): Additional limitations of FFQ = self-report data and potential introduction of recall bias and social desirability bias.
References
Re high number of self-citations: Suggest substituting other references when feasible:
· Self-citation is needed: 12,17,18, 20, 21
· Other references available: 1, 2,3,13, 37
Author Response
This large, prospective cohort study offers unique insights into the potential risk of persistent intakes of UPF and various health conditions. By categorizing the NOVA UPF classification into food groups based on nutritional content and purpose, the researchers evaluated risk based on subtype of UPF—this novel approach contributes unique findings to the field. The ethnically diverse sample allows for generalization of the findings to the Brazilian population, excepting pregnant women since they were excluded from the cohort. Specific suggestions follow.
Comment 1: Keywords. Suggest adding processed meat.
Response:
Thank you for your suggestion. We have now added the keyword.
Comment 2: Introduction. Good overview of topic and introduction to study aim.
Materials and Methods. Are the structured questionnaire (line 96) and CIS-R (line 100) validated tools? If so, note that. If not, add as limitation that these tools were not validated.
Employment of Poisson appropriate given aim to evaluate probability of event given fixed interval of exposure.
Response:
Thank you for your constructive comments. Yes, all the questionnaires used to define the main exposure and outcomes, such as CIS-R, were validated and the corresponding references included (they are now references 18 and 23, respectively). To make this clear we added a somewhat more detailed CIS-R description, also describing its derived outcomes.
We used Poisson regression with robust variance, a frequently used and valid method to generate relative risks in studies with outcomes ascertained at fixed follow-up visits.
Comment 3: Results
Handling of missing data was addressed.
Table 1: Suggest separating the first column into three discrete columns: (1) N, (2) mean, and (3) median.
Response:
Thank you for your suggestion. We have reorganized the table following your suggestion.
Comment 4: Re alcohol consumption, why is it 0 g/day but distilled alcoholic beverages 2.3 g/d?
Response:
Total alcohol consumption includes all alcoholic beverages and is presented in the table as median and interquartile range. Among the subgroups of ultra-processed food are distilled beverages. This consumption, for uniformity with that of the other subgroups, is presented as mean and standard deviation.
Comment 5: Comparison with US cohorts (line 332): Other studies have found that intake of refined grains is not associated with increased risk of T2D when you consider refined grains as a distinct food category. I think this is an important point to make given the study aim. See 10.1016/j.mayocp.2022.05.004.
Response:
We have added a phrase in the Discussion noting this point: “…That a meta-analysis of refined grains suggests a lesser risk than generally assumed for this food group increases the importance of better-characterizing diabetes risk factor [26].”
Comment 6: Numerous reasons for harmful effects (line 361): Recommend including information on theories of how substances in UPFs (additive, preservatives, artificial colors/flavors) may disrupt human physiology. For example, leaky gut hypothesis, induction of Maillard Reaction theory, and cellular inflammation.
Response:
Thank you for your suggestion. We have improved the previous paragraph in an attempt to encompass most of the mechanisms cited in the literature.
Please see our improvements that have been added to the Discussion section: ‘Components of UPFs …, and advanced glycation end products, ….. appear to have harmful effects on the human body [29–32]. Among the potential mechanisms of action of these components are interference with cell signaling pathways involved in glucose homeostasis [32] and alterations in the gut microbiota and gut barrier function, which can lead to inflammation and metabolic changes [33,34]. Intensive processing may also affect digestion, nutrient absorption, and satiety by altering the food matrix [35]’.
Comment 7: Limitations (lines 371-372): Additional limitations of FFQ = self-report data and potential introduction of recall bias and social desirability bias.
Response:
We have added the text: “First, food frequency questionnaires are known to present problems with precision and biased responses.”
Comment 8: References
Re high number of self-citations: Suggest substituting other references when feasible:
Self-citation is needed: 12,17,18, 20, 21
Other references available: 1, 2,3,13, 37
Response:
Most of these references highlight key methodological features of our study in greater detail:
Study design of the ELSA-Brasil baseline:
Aquino, E.M.L.; Barreto, S.M.; Bensenor, I.M.; Carvalho, M.S.; Chor, D.; Duncan, B.B.; Lotufo, P.A.; Mill, J.G.; Molina, M.D.C.; Mota, E.L.A.; et al. Brazilian Longitudinal Study of Adult Health (ELSA-Brasil): Objectives and Design. Am. J. Epidemiol. 2012, 175, 315–324, doi:10.1093/aje/kwr294.
Study design of ELSA-Brasil follow-up:
Schmidt, M.I.; Duncan, B.B.; Mill, J.G.; Lotufo, P.A.; Chor, D.; Barreto, S.M.; Aquino, E.M.L.; Passos, V.M.A.; Matos, S.M.A.; Molina, M. del C.B.; et al. Cohort Profile: Longitudinal Study of Adult Health (ELSA-Brasil). Int. J. Epidemiol. 2015, 44, 68–75, doi:10.1093/ije/dyu027.
Validity of the food frequency questionnaire:
Molina, M.D.C.B.; Benseñor, I.M.; Cardoso, L. de O.; Velasquez-Melendez, G.; Drehmer, M.; Pereira, T.S.S.; Faria, C.P. de; Melere, C.; Manato, L.; Gomes, A.L.C.; et al. Reproducibility and Relative Validity of the Food Frequency Questionnaire Used in the ELSA-Brasil. Cad. Saude Publica 2013, 29, 379–389.
The remaining reference refers to one of the few previous studies investigating UPF food groups.
Canhada, S.L.; Vigo, Á.; Levy, R.; Luft, V.C.; da Fonseca, M. de J.M.; Giatti, L.; Molina, M. del C.B.; Duncan, B.B.; Schmidt, M.I. Association between Ultra-Processed Food Consumption and the Incidence of Type 2 Diabetes: The ELSA-Brasil Cohort. Diabetol. Metab. Syndr. 2023, 15, 1–10, doi:10.1186/s13098-023-01162-2.
We believe that citation of the above references is justified. However, to decrease self-citation, following your suggestion, we excluded one background reference (previous number 1) and one methods reference (previous number 21).
Round 2
Reviewer 1 Report
Comments and Suggestions for Authors
The authors responded to the comments and introduced corrections where possible. I believe that the publication in its current form may be considered for publication.
Reviewer 2 Report
Comments and Suggestions for Authors
I think all responses to reviewers' comments have been addressed satisfactorily.
I have no comments on the revised manuscript.